# Gap Junctional Communication Required for the Establishment of Long-Term Robust Ca^2+^ Oscillations Across Human Neuronal Spheroids and Extended 2D Cultures

**DOI:** 10.3390/cells14211744

**Published:** 2025-11-06

**Authors:** Jasmin Kormann, Eike Cöllen, Ayla Aksoy-Aksel, Jana Schneider, Yaroslav Tanaskov, Kevin Wulkesch, Marcel Leist, Udo Kraushaar

**Affiliations:** 1NMI Natural and Medical Sciences Institute at the University of Tübingen, 72770 Reutlingen, Germany; jasmin.kormann@nmi.de (J.K.); Ayla.AksoyAksel@nmi.de (A.A.-A.);; 2Doerenkamp-Zbinden Chair for In Vitro Toxicology and Biomedicine, University of Konstanz, 78457 Konstanz, Germany; eike.coellen@uni-konstanz.de (E.C.); yaroslav.tanaskov@uni-konstanz.de (Y.T.); kevin.wulkesch@uni-konstanz.de (K.W.)

**Keywords:** Ca^2+^ oscillations, gap junctions, human dopaminergic neuronal-like cells, network synchronization

## Abstract

Synchronized oscillatory fluctuations in intracellular calcium concentration across extended neuronal networks represent a functional indicator of connectivity and signal coordination. In this study, a model of human immature neurons (differentiated from LUHMES precursors) has been used to establish a robust protocol for generating reproducible intracellular Ca^2+^ oscillations in both two-dimensional monolayers and three-dimensional spheroids. Oscillatory activity was induced by defined ionic conditions in combination with potassium channel blockade. It was characterized by stable frequencies of approximately 0.2 Hz and high synchronization indices across millimeter-scale cultures. These properties were consistently reproduced in independent experiments and across laboratories. Single-cell imaging confirmed that oscillations were coordinated throughout large cell populations. Pharmacological interventions demonstrated that neither excitatory nor inhibitory chemical synaptic transmission influenced oscillatory dynamics. Gap junction blockers completely disrupted synchronization, while leaving individual cell activity unaffected. Functional dye-transfer assays provided additional evidence for electrical coupling. This was further supported by connexin-43 expression profiles and immunostaining. Collectively, these findings indicate that synchronized Ca^2+^ oscillations in LUHMES cultures are mediated by gap junctional communication rather than by conventional synaptic mechanisms. This system offers a practical platform for studying fundamental principles of network coordination and for evaluating pharmacological or toxicological modulators of intercellular coupling. Moreover, it may provide a relevant human-based model to explore aspects of neuronal maturation and to assess compounds with potential neurodevelopmental toxicity.

## 1. Introduction

Calcium (Ca^2+^) signaling, characterized by tightly regulated intracellular Ca^2+^ fluctuations, is essential both for healthy neurodevelopment and mature neuronal functioning. These Ca^2+^ dynamics activate various signaling cascades, crucial for neural network formation and communication [1]. A special form of intracellular Ca^2+^ dynamics is oscillations, i.e., a periodic up- and down-regulation of the concentration of free intracellular Ca^2+^ in the whole cell or parts of a cell. This phenomenon has been observed in various neuronal cells and culture conditions [2,3]. Calcium oscillations in single cells can be spontaneous or be triggered by external stimuli, with their frequency encoding the strength of a stimulus (e.g., the extracellular concentration of a transmitter). However, there are also oscillations across large cell networks (such as various forms of brain waves with high (50–80 Hz) to low (<1 Hz) frequencies. It is firmly established that in the brain, a high degree of synchronization between many neurons can often be found [4,5,6]. Notably, oscillations of calcium levels do not necessarily correspond to synchronized action potentials (APs). Many APs may occur during one oscillation period; alternatively, there may also be a lack of APs for several periods. Thus, the oscillation of Ca^2+^ levels is not just a reflection of neuronal electrical signaling, but it may be modulated or even mainly triggered by metabolic processes in the cell (e.g., loading of organellar stores or phosphorylation levels of proteins) [7,8,9]. Accordingly, aberrant patterns of Ca^2+^ oscillations have been observed in epileptic seizures or after stroke [10,11], suggesting a crucial role of oscillations in neuronal networks in normal and diseased brain functions.

Several hypotheses are being discussed about the mechanisms involved in the generation and coordination of oscillatory activity in the brain during neurodevelopment and in later life [7]. As chemical synapses are the most important key players in adult neuronal signaling, it is assumed that they also play a role in the immature nervous system. However, electrical synapses, based on gap junction coupling, could contribute to the maintenance of the oscillatory synchronization during neurodevelopment [12,13]. Gap junctions are pores composed of interconnected connexin (Cx) protein subunits in hexameric arrays within the plasma membranes of two adjoining cells. They can be found in many tissues, in which rapid signal transfer in the form of ions and/or neurotransmitters is crucial [14,15]. They are critical during neurodevelopment, mediating proliferation and differentiation processes, and play an essential role in the formation of the first neural circuits before chemical synapses are established [14,16]. Moreover, the importance of gap junctions is also evident from mutations in the connexin genes that result in severe diseases, such as Charcot–Marie–Tooth disease or non-syndromic deafness [17,18,19,20].

The involvement of electrical synapses during human neurodevelopment is relatively understudied. Moreover, human cell-based models that develop electrical synapses and that allow capturing their function at a reasonably high throughput are scarce.

Here, Ca^2+^ oscillatory behavior was studied in a well-established human dopaminergic neuronal model. The LUHMES precursor cells used for this are kept in a proliferative state by a conditionally active ectopic gene (*v-myc*). Upon inactivation of the transgene (by tetracycline), the cells initiate a neuronal differentiation process and become fully and irreversibly post-mitotic [21]. They form an intricate neurite network, express a large panel of pre- and post-synaptic proteins (including channels, receptors, and transporters relevant for neurotransmission), and they generate spontaneous and evoked APs [21,22,23,24]. They have been extensively used to study certain aspects of neurodegenerative diseases, such as protein aggregation of mitochondrial dysfunction [25,26], and they have been used as a platform for neurotoxicology screenings [27,28,29,30,31,32]. As for any model system, one has to be aware of the limitations of LUHMES cultures. The cells resemble rather immature neurons (e.g., indicated by a residual doublecortin expression), and there has not been any convincing demonstration that they can form synapses with one another. In line with the likely absence of chemical synapses, the dendritic arborization is quite limited, and there is no evidence for expression of spines. However, the cells have proved to be very useful for studying certain Ca^2+^ signaling pathways (e.g., triggered by nicotine), and there is some evidence that LUHMES cultures exhibit synchronous Ca^2+^ oscillations when exposed, e.g., to amphetamine [23]. For this reason, we refer to the cells here as “LUHMES cultures” or “LUHMES neuron-like cells” to distinguish them from primary adult neurons as found in brain tissue.

In this study, we explored the usefulness of LUHMES cultures to establish a novel model of Ca^2+^ oscillations in human neuronal-like cells. After characterization of the Ca^2+^ spiking pattern in two-dimensional monolayers, three-dimensional spheroids were also generated to measure the oscillation pattern at various differentiation time points. Data from a combination of phenotyping and functional studies, combined with pharmacological intervention experiments, were obtained to determine the role of gap junctions. Finally, we aimed to characterize and optimize this novel test system to allow future laboratory transfers and a screening campaign to identify factors affecting neuronal coordination and thus neurodevelopmental functions.

## 2. Materials and Methods

### 2.1. LUHMES Cell Culture

LUHMES (Lund human mesencephalic) cells are a subclone of the mesencephalic cell line, which was derived from ventral mesencephalic tissue of an 8-week-old female human embryo. The cells (ATCC number: CRL-2927) have a normal karyotype and a typical primary gene sequence, except for a conditionally active *v-myc* transgene [33] with a known insertion site. They have been short tandem repeat (STR) authenticated [34]. LUHMES cultures were cultured following established protocols [21,35,36]. Briefly, cell culture flasks were pre-coated with 1 µg/mL fibronectin (Merck KGaA, Darmstadt, Germany) cultures were maintained in Advanced DMEM/F12 (Dulbecco’s Modified Eagle Medium/Ham’s F-12) (Gibco, New York, NY, USA) supplemented with 2 mM L-glutamine (Merck KGaA), 1x N2-supplement (Gibco), and 40 µg/mL fibroblast growth factor (FGF) (R&D Systems, Minneapolis, MN, USA). Cells were passaged upon reaching approximately 70% confluency. For passaging, cells were washed once with phosphate-buffered saline (PBS), detached using 0.05% trypsin-ethylenediaminetetraacetic acid (EDTA) (Gibco), which was incubated for 1 min at 37 °C with 5% CO_2_, resuspended in medium, and centrifuged at 300× *g* for five min at room temperature. For differentiation, cells were cultured in advanced DMEM/F12 supplemented with 2 mM L-glutamine, 1 × N2-supplement, 1 mM N^6^,2′-O-dibutyryl 3′,5′-cyclic adenosine monophosphate (cAMP) (Merck KGaA), 1 µg/mL tetracycline (Merck KGaA), and 2 ng/mL glial cell line-derived neurotrophic factor (GDNF) (R&D Systems). After two days of differentiation in PLO/fibronectin-coated flasks, cells were transferred to the final culture dish. The detachment process followed the previously described protocol, with the modification that trypsin was incubated for 5 min. For two-dimensional cultures, cells were seeded at a density of 200,000 cells per cm^2^ onto culture plates pre-coated with 0.1% polyethylenimine (PEI) (Merck KGaA). To prepare the coating, 0.1% PEI was added to the wells and incubated for 1 h at 37 °C with 5% CO_2_. After incubation, the coating solution was removed, and the wells were washed four times with double-distilled water. The water was then removed, and the plates were allowed to dry under a sterile hood. Half-media changes were performed every other day, and monolayer cultures were recorded on day 9 of in vitro differentiation (DIV 9). For three-dimensional cultures, LUHMES cells were cultivated using protocols established earlier [37]. Briefly, cells were seeded into 96-well ultra-low attachment plates (BIOFLOAT™, Sarstedt, Nümbrecht, Germany) at a density of 7500 cells per well, following a two-day differentiation process in flasks (as described for 2D cultures). The plates were centrifuged at 290× *g* for 3 min to facilitate spheroid formation. Medium changes were performed every other day. On the sixth day, spheroids were transferred to plates coated with Matrigel (Merck KGaA; 1:25 in medium). Cultivation continued for two more days, and spheroids were analyzed/recorded on DIV 10.

### 2.2. Ca^2+^ Imaging

High-throughput Ca^2+^ imaging recordings were performed using the FDSS/µCELL (Hamamatsu Photonics, Hamamatsu City, Japan). Intracellular Ca^2+^ changes were visualized using the Ca^2+^ indicator Cal-520, AM (AAT Bioquest, Pleasanton, CA, USA) with cells stained at 1 µM for two-dimensional cultures and 2 µM for three-dimensional cultures. The Ca^2+^ indicator dye was incubated for 1 h at 37 °C and 5% CO_2_. The system records the mean fluorescence intensity across the entire well over the recording period. An underside camera captures images of up to 384 wells simultaneously. At the same time, a dispensing head applies up to 384 different compounds at once, allowing the simultaneous measurement of compound effects across the entire plate. Compounds were applied as a 1:10 dilution to achieve the final concentration in the wells. Recordings were performed in a buffer containing [in mM]: 135 sodium chloride, 1 potassium chloride, 10 D-glucose, 0.2 magnesium chloride, 10 HEPES, and 3.8 calcium chloride. The pH was adjusted to 7.4, and the osmolarity was set to 300 ± 5 mOsm. Compounds were applied after a 2.5-min baseline period. A second compound application was performed after another 5 min. For Ca^2+^-imaging recordings, Greiner 384-well μClear™ microwell plates (Greiner Bio-One GmbH, Frickenhausen, Germany) were used. Unless stated differently, all chemicals were purchased from Merck KGaA.

The FDSS/µCELL system has limited resolution, preventing the visualization of single-cell activity or the activity of subpopulations. To achieve higher resolution, a Nikon Ti2 Eclipse system (Nikon Instruments, Amstelveen, Netherlands) with an Andor iXon Life 897 EMCCD camera (acal bfi, Dietzenbach, Germany) was utilized. Cell handling, recording buffer, and calcium dye preparation were conducted as described above.

### 2.3. Patch Clamp Recordings

For patch clamp recordings, LUHMES cells were seeded onto glass coverslips at a density of 1500 cells/µL in 20 µL droplets. The coverslips were pre-coated with 0.1% PEI solution. In brief, the PEI coating solution was added to the glass coverslips and incubated for 1 h at 37 °C and 5% CO_2_. The coverslips were then washed with double-distilled water four times and allowed to dry under the sterile hood. GB150TF-10 glass pipettes (Science Products, Hofheim, Germany) were prepared on the day of recording using a micropipette puller (Sutter Instrument P-1000, Novato, CA, USA) and had a resistance of 3–4 MΩ. Electrophysiological recordings were performed using an EPC 10 USB patch clamp amplifier in combination with the PatchMaster software (version 2 × 90.5; HEKA Elektronik, Lambrecht, Germany). The internal solution consisted of [mM]: 107 K-gluconate, 10 KCl, 1 MgCl_2_, 10 HEPES, 5 EGTA, 4 Na_2_ATP, and 0.2 NaGTP. The pH was adjusted to 7.2, and the osmolarity was set to 280 ± 5 mOsm. The external solution consisted of [mM]: 140 NaCl, 4 KCl, 1 MgCl_2_, 1.8 CaCl_2_, 10 HEPES, and 10 D-glucose. The pH was adjusted to 7.4, and the osmolarity was set to 300 ± 5 mOsm. All chemicals were purchased from Merck KGaA.

### 2.4. Dye-Transfer Experiments

For dye transfer experiments, 50 µM Alexa-488 (Life Technologies, Carlsbad, CA, USA) was loaded into a recording pipette. The internal and external solutions, as well as the patching protocol, were identical to those used for patch clamp recordings, apart from the Alexa-488 dye. Cells were patched in whole-cell mode [38]. The dye was allowed to distribute within the cell over a 15-min period. The dye was injected by the application of hyperpolarizing current pulses of 1 nA for 200 ms. In some experiments, 30 µM 18β-Glycyrrhetinic acid (18β-GA) (Merck KGaA) was applied to the bath solution 15 min before patching the cell with the Alexa-488-filled recording pipette.

### 2.5. Immunohistochemistry

Immunohistochemistry was performed using an adapted protocol from Scholz and colleagues [21]. LUHMES cells were seeded at a density of 215,000 cells per well (207,000/cm^2^) onto coated eight-well ibidi slides (Ibidi GmbH, Gräfelfing, Germany). Wells were coated with 300 μL of an adapted coating solution containing PLO [100 μg/mL] and fibronectin [2 μg/mL] in Milli-Q water. On the day of seeding, wells were washed twice with 300 μL PBS. Cells were cultivated until differentiation at d0, d3, and d9. Cell culture medium was removed and replaced with 10% paraformaldehyde (PFA) for fixation at room temperature for 15 min. Cells were then rinsed once with PBS. Permeabilization was carried out using permeabilization buffer for IHC (0.6% Triton X-100 in PBS) for 10 min at room temperature, followed by a PBS wash. For blocking, cells were incubated in IHC blocking buffer (5% FCS, 0.1% Triton X-100 in PBS) for 1 h at room temperature. Primary antibodies, rabbit anti-connexin 43α serum (1:400), mouse IgG2 anti-TUJ1 (1:1000), and anti-tyrosine hydroxylase clone LNC1 (1:200), were diluted in blocking buffer for IHC and incubated overnight at 4 °C. After incubation, cells were washed once with PBS. Secondary antibodies, Alexa FluorTM 488 chicken anti-rabbit IgG (H+L), Alexa FluorTM 647 goat anti-mouse IgG2a (y2a), and Alexa FluorTM 555 goat anti-mouse IgG1 (y1), all diluted 1:1000 in blocking buffer, and Hoechst-33342 (1:1000 in blocking buffer) were added for 30 min at room temperature in the dark. Finally, cells were washed once with PBS. Unless stated differently, all chemicals were purchased from Merck KGaA.

Imaging was performed using a Zeiss LSM880 confocal microscope (Zeiss, Oberkochen, Germany) with either a 40×/1.40 PlanApochromat (Oil) objective or a 63×/1.40 PlanApochromat (Oil) objective (Zeiss), utilizing the appropriate fluorescence channels corresponding to the respective fluorophores. Multiple images were captured per well, with focus individually adjusted for each structure. Images were exported in TIFF format. Qualitative analysis was conducted in ImageJ Fiji (version 1.54p), with adjustments to brightness and contrast applied solely for visualization purposes. The inserted scale bars were 20 μm.

### 2.6. Transcriptomics

LUHMES cells were seeded at a density of 60,000 cells/well (207,000 cells/cm^2^). Every other day (d), 50% of the medium was exchanged. Samples were taken on d0, d1, d2, d4, d6, d8, and d9. For sample preparation, the medium in 96-well plates was replaced by BioSpyder lysis buffer (33 µL/well; BioSpyder Technologies, Glasgow, UK). Following a 10-min incubation at 37 °C and 5% CO_2_, the plates were sealed and frozen at −80 °C for lysis completion.

Targeted transcriptome sequencing (including QC, alignment, and read quantification) was conducted at Bioclavis (BioSpyder Technologies) using the TempO-Seq technology. For each of the targeted genes, a 50 bp fragment was amplified, while also introducing sample-specific barcodes, which subsequently enabled sample pooling for the next-generation sequencing of the collection. A reference library containing the collection of all amplification products was used for assigning read counts to each targeted gene. A pre-filtering step for library size (<0.2 million) and average gene count (<1.5) was performed. The counts per gene were normalized to counts per million by dividing by the total number of mapped reads per sample and multiplying by 10^6^. The effect of normalization was internally checked, and no outlier samples were identified. A gene set list of interest was defined to account for all gap junction genes, and expression values were obtained for these genes. For the heatmap generation, the mean counts of the specified gene list were used. The mean counts for each differentiation day (dx) are shown as log2 values indicated by color, blue for low log2 values and red for high log2 values.

### 2.7. Data Analysis

Data were analyzed using either Python [39] or RStudio [40] based scripts. 

In this study, the global synchronization index (GSI) was used to assess the synchronization of the culture. The GSI was computed using Python-based scripts (v7.0.8) (courtesy of Emilio Pardo-Gonzalez) and was initially introduced by Li et al. [41] to quantify neuronal synchronization across multiple sites (unlike other methods that compare only two time series). This approach constructs a correlation matrix for all pairs of recorded signals. For each imaging series within a single well, the activity of 10 individual regions of interest (ROIs) was correlated. Fluorescence data were extracted using ImageJ (FIJI). This correlation matrix was then used in an eigenvalue-based algorithm to determine the GSI from the recorded data. The GSI ranges from 0 to 1, where 0 indicates non-coordinated activity, and 1 signifies a highly interconnected network of cells [41].

The following packages were used for the computation of the GSI: Pandas [42], NumPy [43], Matplotlib [44], and Seaborn [45]. 

All other graphs were generated using RStudio (v4.2.3). For frequency analysis, a fast Fourier transformation (FFT) was performed using the genecycle package [46,47]. The function periodogram from the same package was used to compute the power spectral density and Fourier frequencies. The package incorporates the approaches originally presented in other works [46,48,49]. 

The following R-packages were used to generate the figures: ggplot2 [50], dplyr [51], broom [52], drc [53], cowplot [54], grid [55], multicomp [56], magrittr [57], plotrix [58], patchwork [59], ggbeeswarm [60], readr [61], readxl [62], genecycle [46,47], ggsignif [63], purr [64]. 

### 2.8. Data Handling and Statistics

All data sets include at least three repeats. Unless stated otherwise, error bars represent the mean ± SEM. A *p*-value of <0.05 was considered statistically significant. All data presented in manuscript figures are available in Excel files, such that other displays or statistical approaches may be applied to them (Appendix A).

## 3. Results 

### 3.1. Synchronous Ca^2+^ Oscillations

Highly synchronized Ca^2+^ oscillations have been observed in LUHMES cultures under some specific pharmacological stimulation conditions [23]. To follow-up on this, we explored culture conditions that would favor oscillations (i) that are highly reproducible (ii) coordinated over long (mm dimension) ranges, (iii) stable over extended periods (>15 min) (iv) not requiring fine-tuning of exact concentrations of a potent pharmacological agent, (v) and being suitable for neurobiological and toxicological characterization. We found that cultivating LUHMES cells in a high extracellular calcium [3.8 mM] and low potassium [1 mM] environment favors their oscillation tendency. On this basis, long-lasting synchronization was reproducibly triggered by the addition of tetraethylammonium chloride (TEA) [20 mM], a known blocker of a large variety of potassium channels [65,66,67,68]. 

To make sure that only signals from long-range synchronized Ca^2+^-changes were observed, we used an imaging system (FDSS/µCELL) that records the average fluorescence signal from an entire culture in each well of a 384-well plate (producing one single fluorescence value per time point and well). Because asynchronous single-cell activity averages out at the population level, a regularly oscillating well-average signal is indicative of coordinated (phase-aligned) activity across many cells; we therefore interpret robust well-average oscillations as evidence of network-level synchronization. Based on this, we interpreted regular oscillations of Ca^2+^ as an indication that cells from all locations within the well had a synchronized activity, i.e., that activities of individual cells would not result in a relatively constant average sum (Figure 1A).

We observed exactly this functional response: Following TEA addition, a sharp rise in average intracellular free Ca^2+^-concentrations (shortly called “Ca^2+^ levels” in the following text) was observed. This was immediately followed by synchronous and highly reproducible Ca^2+^ oscillations around a relatively constant average value, which was clearly increased relative to the original baseline. Oscillations persisted for at least 15 min (sometimes for an hour or longer) (Figure 1B).

To obtain more detailed data on the oscillations and their robustness, we analyzed the oscillation frequency across multiple wells from five independent experiments (conducted over a period of more than one year). The oscillatory activity demonstrated remarkable reproducibility across experiments and wells, with a frequency of 0.200 ± 0.007 Hz (*n* = 20 experiments, Figure 1C).

After a clear indication of Ca^2+^ oscillations obtained by whole-well imaging, it was essential to confirm these results by single-cell imaging. Such an approach is not only an essential experimental verification, but it also allows for obtaining more detailed information about the propagation of the signal throughout the culture. The experimental protocol was essentially similar, but the fluorescence signal was recorded by a high-resolution camera mounted on an inverted microscope. Thus, the fluorescence of ROIs was recorded in parallel within one culture. First, a large ROI was used, which corresponded to the entire imaging field. A clear oscillation of the signal was observed (Figure 2A). This control experiment showed that the alternative recording on two imaging devices showed essentially similar results. Next, several ROI, corresponding to the positions of selected cell bodies, were defined. An oscillating signal was observed for all cells, and oscillations occurred synchronously (as by visual inspection) (Figure 2B). Note that the duration (half-amplitude width) of Ca^2+^ transients in whole-field and soma ROIs reflects Ca^2+^ handling and indicator/detection kinetics and cannot be used to infer the duration of underlying electrical depolarizations; oscillation frequency was quantified independently by FFT (0.200 ± 0.007 Hz).

To obtain a quantitative measure of synchronicity, we assessed network coordination by calculating the GSI for ten randomly selected ROIs across the entire wells (Figure 2C,D). The two-dimensional culture exhibited a GSI of 0.84 ± 0.02 (n = 9 experiments), indicating a highly interconnected network.

### 3.2. Oscillatory Activity in a 3D Model

Given the complex three-dimensional architecture of the human brain, we next evaluated whether oscillatory activity could also be observed in a 3D spheroid model, based on the same cells. Large lots of LUHMES spheroids were generated in suspension cultures [37]. For experiments, single spheroids were transferred to imaging plates and left to adhere for two days. Using the same induction protocol as for the two-dimensional system, robust and long-lasting oscillations were successfully elicited in the 3D spheroids (Figure 3A–D).

The oscillation frequency was quantified for 20 spheroids from five independent differentiations, collected over a period of more than one year. The 3D cultures exhibited a frequency of 0.200 ± 0.004 Hz (*n* = 20). Thus, the mean frequency in spheroids was highly similar to that recorded for 2D cultures. This provides evidence for a high reproducibility and consistency of the model across formats (Figure 3E,F).

To assess network synchronization, the GSI was calculated across multiple spheroids, using ten randomly selected ROIs per spheroid. The 3D cultures demonstrated a GSI of 0.89 ± 0.02 (*n* = 9), reflecting a high degree of coordinated activity within a given spheroid similar to 2D cultures (Figure 3G).

### 3.3. Transferability and Viability Control of the Model System

For later applications of the novel model system described above, e.g., for toxicity testing and basic mechanistic research, it is crucial that the method can be run in other laboratories, by other operators, and possibly using other analytical instrumentation [69,70,71,72,73]. This was tested by a transfer between the NMI in Reutlingen (see affiliations) and the University of Konstanz. Two different imaging devices were used at the recipient laboratory, and two different operators performed the experiments there over a period of one year. The stable and long-lasting oscillations were consistently reproduced in two-dimensional cultures (Figure 4A–C) but also in 3D spheroids (Figure 4D). In this experimental setting, it was also possible to measure oscillations in the neurites (Figure 4E). The oscillations showed similar quantitative characteristics in both labs (frequency, synchronization) (Figure 4G,H).

Having considered reproducibility issues, the next step was to verify that oscillations did not primarily indicate an initiated cell death process or a catastrophic failure in energy supply. We found that cellular ATP levels were fully maintained for 2.5 h in cells triggered to undergo oscillations after a change of buffer and an addition of TEA. As a positive control for this initial viability measure, we simultaneously blocked the mitochondrial ATP synthase (with oligomycin [1 µM]) and glycolysis (by the glucose transporter inhibitor Glutor [100 nM]). Under such conditions, ATP was depleted by 80% after 10 min, and completely after 20 min. The latter data indicate that LUHMES cultures have a continuous high ATP turnover, and that the levels would rapidly and steeply fall as soon as new production is impaired. Indirectly, these data indicate that oscillating cells still produced substantial amounts of ATP (for at least 2.5 h) that maintained the cellular levels despite a continuous consumption (Appendix A).

We also measured the viability of spheroids using calcein-AM [1 µM] (staining only live cells). No difference was observed after a 24 h treatment in normal medium compared to oscillation buffer (containing TEA) (Appendix A). As a third approach, we measured mitochondrial respiration (oxygen consumption rate, OCR) of oscillating spheroids. This was maintained similarly to controls (in normal medium) during the normal period of our experiments (15 min). Extended measurements (2.5 h) indicated a continuous decline of mitochondrial respiration (OCR) to about 50%. However, this decline was compensated for by a similar time course by an increase in glycolysis (Appendix A). These findings suggest that the maintained ATP levels were a result of a partial shift in energy production from mitochondria to glycolysis. This would be in line with a partial dissipation of the mitochondrial membrane potential by a massive uptake of Ca^2+^ (being transported along the electrochemical gradient across the mitochondrial inner membrane).

In summary, these data suggest that neither the cell membrane nor the cellular ATP levels nor the mitochondrial respiratory activity was impaired during our oscillation experiments. Moreover, the data suggest that some metabolic shifts are occurring in the oscillation buffer, which may contribute to the oscillations.

### 3.4. Continuation of Oscillatory Activity in the Presence of Strong Modulators of Chemical Synapses

Besides some potential intracellular changes that affect [Ca^2+^]_i_, there must be a mechanism that is responsible for the coordination between individual cells. The most established mechanisms for this in the nervous system are chemical and electrical synapses [5,6,74,75,76,77]. We first examined the contribution of chemical synapses, as they play the dominant role in the mature nervous system [78,79,80]. As glutamate is by far the most important excitatory neurotransmitter in the human brain, the effect of blocking glutamate receptors was studied: we applied CNQX to block AMPA receptor-mediated signaling and DL-AP5 to inhibit NMDA receptor activity. To further evaluate the overall contribution of glutamatergic transmission, the effect of exogenous glutamate application was tested. The oscillatory activity remained unaffected by any of these interventions. A quantification of the oscillation frequency showed that none of the compounds (CNQX, DL-AP5, and glutamate) had a significant effect (Figure 5A,B and Appendix A).

We additionally examined the role of the GABAergic system in regulating oscillatory activity. Picrotoxin was used to block GABA_A_-receptors, and phaclofen to inhibit GABA_B_-receptors. To assess the effect of GABA as an inhibitory neurotransmitter, exogenous GABA was also applied. Similar to the glutamatergic manipulations, none of these treatments affected the oscillation frequency, which remained at approximately 0.2 Hz (similar to control conditions) (Figure 5C).

### 3.5. Expression and Role of Gap Junctions

Gap junctions form direct electrical connections between adjacent cell membranes, enabling the passage of electrical signals [14,81]. We next explored whether electrical synapses play a role in the observed oscillatory activity in LUHMES cultures. To this end, we applied two different gap junction blockers, carbenoxolone (CBX) and octanol, and assessed whether gap junctional communication and connectivity are present in LUHMES neuronal-like cells. Both compounds completely abolished the oscillatory activity, indicating a critical role of electrical synapses in mediating network synchronization (Figure 6A–C). A subsequent FFT analysis of four recordings after treatment with gap junction blockers showed that there was no clear frequency peak at all. These data suggest that there was no synchronous pattern of Ca^2+^ oscillations under these conditions (Figure 6D). These experiments were also reproduced in a second laboratory to make sure that the observed phenomena (similar in both laboratories) were also due to the same mechanistic basis (Figure 4H).

We wanted to provide an additional line of evidence for the existence of functional gap junctions between LUHMES cells, independent of the use of pharmacological modifiers. For this reason, we examined the transfer of a low molecular weight, membrane-impermeable fluorescent dye (Alexa-488) from an injected cell towards its non-manipulated neighbor cells. Following loading of a single cell in a d4 culture with a dye-filled patch clamp pipette, we observed the spread of the fluorescence signal to about 5–17 (5 cells *n* = 1; 6 cells *n* = 1; 7 cells *n* = 1; 9 cells *n* = 1; 11 cells *n* = 1; 17 cells *n* = 1) neighboring cells. The gap junction inhibitor 18β-GA blocked dye-transfer at d4 completely in five experiments and nearly completely (one neighboring cell stained) in one experiment (Figure 7A). Dye spread was also observed at later stages of differentiation, but only to one or two neighboring cells (1 cell *n* = 5; 2 cells *n* = 1). In d9 cells, 18β-GA completely abolished dye transfer in all experiments (*n* = 6) (Figure 7B). These data suggest that there are gap junctions between LUHMES neuronal-like cells that allow intercellular transfer of small molecules. The peak of gap junctional connectivity appears to be at earlier stages of differentiation. However, it is maintained until at least d9, and may therefore account for the observed Ca^2+^ oscillations.

As a further line of evidence, we obtained gene expression data for gap junction constituents over the time course of LUHMES differentiation. Expression of the GJA1 gene, encoding the major gap junction protein connexin 43, was particularly high. The maximum levels were observed in immature neuronal-like cells, but substantial expression was also observed in more mature (d9) neuronal-like cells used here as a model system (Figure 6E). These results are fully in line with the functional dye transfer data. 

### 3.6. Identification of Gap Junctions Between Adjacent LUHMES Cell Membranes

To investigate whether the gene expression results translate to protein levels, immunocytochemistry for connexin 43 was performed on LUHMES cultures. ßIII-tubulin was selected as a marker for the visualization of neurites, enabling the localization of connexin 43α to be detected. The staining intensity was most pronounced for d0 cells. It slowly decreased throughout the differentiation, but was still clearly visible at d9. This is in concordance with the results from the gene expression analysis. The mean counts of the specified gene list can be found in Appendix A.

A dot-like staining was observed at the contact points of cell membranes. Gap junctions at such positions may support the formation of electrical synapses between neighboring cells. The signal was not only located on the somata of the cells but also on their neurites, indicating that electrical coupling to neighboring cells may involve neurites, in addition to the somata (Figure 4D and Figure 8). The studies on gene expression, protein localization, and dye coupling all suggested that gap junctions, as a basis for oscillatory activity, were already present in immature (early stage differentiation) cultures. Therefore, we also investigated whether early-stage cells already exhibited oscillatory activity. Indeed, oscillations were observed already early in the differentiation process (Appendix A). These results are in line with other findings that suggest a particularly important role of gap junctions in the electrical coupling of neuronal networks at early developmental stages [16,82,83]. For instance, it has been shown that gap junctions can propagate changes in Ca^2+^ levels rapidly and directly to neighboring cells [84,85]. The data presented here also agree with findings that connexin 43 is prominently expressed in the nervous system [86,87,88].

### 3.7. Inhibition of Gap Junctions Stops Network Connectivity, but Not Single Cell Activity

We hypothesized that gap junctions primarily mediate the synchronization of network activity, rather than being necessary for the intrinsic activity of single cells (as suggested in another model [12]). To obtain experimental evidence, Ca^2+^-imaging was performed with single-cell resolution in the presence or absence of CBX (to block gap junctions). The response of randomly selected cells was analyzed before and after the addition of CBX to assess changes in synchronization. Under control conditions, the cells exhibited highly synchronized activity. Following the addition of CBX, this synchronicity was disrupted. However, some individual cells continued to display some recurrent Ca^2+^ spikes (mostly non-synchronized and not at one given frequency) (Figure 9A,B). Synchrony was lost after 30–60 s, which probably corresponds to the time until CBX took effect. The GSI was artificially elevated when that transition period was included in the calculation. We therefore excluded the transition period from the calculation of the GSI in order to obtain the actual results (Appendix A). To quantitatively assess this desynchronization, we calculated the GSI before and after drug application for the same randomly selected ROIs (corresponding to single cells). Prior to drug application, the GSI was 0.84 ± 0.02 (*n* = 9), indicating strong network synchronization. After the addition of CBX, the GSI dropped significantly to 0.06 ± 0.03 (*n* = 9), reflecting a substantial loss of coordinated activity across the culture without loss of individual cell activity (Figure 9C–E).

As an alternative technical approach, we performed current-clamp recordings using the same (as for imaging) high-calcium, low-potassium buffer in combination with TEA [20 mM] to induce oscillatory activity. The cells showed spontaneous activity as evidenced by frequent spikes (action potential-like depolarization of the cell membrane). In between the spikes, small depolarization events consistent with spikelet activity were identified. Spikelets are spike-like responses with amplitudes below 30 mV and are typically indicative of electrical coupling via gap junctions [89,90,91]. They were observed in 10 out of 13 recorded cells (Figure 9F,G). Spikelets are known in electrophysiology as the result of an action potential that originates from an electrically connected cell, whose signal is filtered and transmitted via gap junctions [92,93,94]. We found that spikelet activity was almost completely abolished following the application of the gap junction blocker 18β-GA. This suggests that electrical coupling via gap junctions plays a key role in spikelet generation. This implies, at least indirectly, that electrical coupling of LUHMES cells was also observed by an electrophysiological approach. The generation of spikes (cell activity pattern) was also strongly decreased (remaining activity was only recorded in 3 out of 12 cells) after application of 18β-GA (Figure 9H). These electrophysiological data suggest that “spontaneous activity” of the cells was, at least in part, due to their coupling within an oscillating network.

## 4. Discussion

We presented here a novel experimental model that allows a reproducible and quantitative assessment of Ca^2+^ oscillatory activity in LUHMES neuronal-like cells. The consistent reproduction of the data in two independent laboratories, in different culture formats, and on different imaging systems provides evidence for the reliability of the method. The highly synchronized Ca^2+^ oscillations occurred at the same frequency in both 2D and 3D cultures. Pharmacological activation (via glutamate) or inactivation (via inhibition of AMPA and NMDA receptors) of excitatory synapses did not change the pattern of oscillations. Also, the activation or block of inhibitory (e.g., GABAergic receptors) synaptic transmission had no effect. This indicates an alternative mechanism to the classical chemical transmission. Indeed, our findings suggest that synchronous Ca^2+^ oscillations may occur via gap junctions. 

A main finding of our study is that LUHMES cells are functionally coupled via gap junctions and that such cell–cell connections are crucial for synchronous Ca^2+^ oscillations. We not only identified this network feature in our LUHMES cell model, but we also established a high-throughput assay that is suitable for studying such complex neuronal-like network functions and their disturbances. Our findings on the role of connexins in activity patterns of immature neurons are consistent with observations in the developing human cortex, where pharmacological inhibition of connexin pores also markedly suppressed spontaneous depolarizing activity in subplate neurons, indicating a connexin-based mechanism in immature neuronal networks in human brain tissue [95].

We had previously observed the induction of synchronized Ca^2+^ oscillations in LUHMES cultures. The neuronal network activity was activated in a controlled manner, for example, upon dopamine application or modulation of the dopamine transporter [23]. We originally assumed that special conditions involving neurotransmitters would be required for synchronized oscillatory network activity. Our present study indicates that long-term stable oscillations can be induced by simple shifts in ion balances. To understand such a system, it may be useful to define some of its key aspects and then to address them in future studies. Such aspects are: (i) what is the original trigger of the observed activity in LUHMES cultures; (ii) is there a need to repetitive triggers to maintain the observed oscillations; (iii) what regulatory processes are involved in the phases of Ca^2+^ level reduction and Ca^2+^ level increase during each oscillation period; (iv) are these mainly electrophysiological or metabolic processes; (v) what leads to communication across the network to enforce synchronicity; and (vi) how does the measured readout (free cytosolic Ca^2+^ concentrations) correlate with other readouts of neuronal activity? The dissection of the described model system helps to clarify aspects of follow-up work, sharpens the discussion of the findings, and helps to draw connections to other findings.

LUHMES cultures are relatively homogeneous, allowing for a more precise mechanistic interpretation than in more heterogeneous neuronal culture systems, where many cell types influence electrical output. In LUHMES cell networks, it is unlikely that the synchronized Ca^2+^ oscillations are mediated by the release of synaptic neurotransmitters, as to our knowledge, no functional synapses are present in LUHMES cultures. Thus, mechanisms related to synaptic plasticity in other cell lines or to synaptic neurotransmitter release, as described in tissue models [96,97,98], are not relevant explanations for the Ca^2+^ oscillations observed here. In this context, it is also important to recall that most evoked depolarization spreadings in tissue models take milliseconds and end then, if no new stimulus is applied. In contrast, oscillations in the LUHMES model continued for 15 min and longer after a single trigger. Phenomenologically, this more resembles, e.g., delta waves in the brain during certain sleep phases, or oscillations in the developing retina [99,100], although it is unclear whether similar mechanisms apply.

After characterizing the oscillations in 2D and 3D, we focused on one of the six key questions above: this study presents several lines of evidence that electrical synapses, consisting of gap junctions, play a key role in the coordination of the network. Thus, one application of the experimental system described here is its use for the study of gap junction communication in a relevant and well-established model without input and interference from synaptic mechanisms.

While gap junctions are necessary for the oscillations to occur, we have not answered the question of whether they are sufficient alone to trigger oscillations or whether they are the main drivers. We have shown that metabolic changes are associated with the oscillations, but more work is required to probe the causality of candidate mechanisms, such as calcium-induced calcium release from the endoplasmic reticulum, store-operated calcium channels, or fluctuations of mitochondrial membrane potential [7]. The similarity of network oscillations that we observe in different culture formats suggests some cell-intrinsic mechanism. Thus far, the mechanistic basis for this is unknown. A potential explanation for this may be exhaustion of energy stores as described for the pancreatic β-cell [101,102] or dissipation of ion gradients that require time to be restored, before the next oscillation spike can occur. This potential intracellular mechanistic basis was beyond the scope of the present study, but it certainly requires attention in the future.

In our model, it is more likely that the specific ionic conditions of the buffer in combination with TEA-induced depolarization provide the necessary electrophysiological state initiating and maintaining synchronized Ca^2+^ oscillations. However, several questions remain open. We do not know whether the initial trigger may depend on pacemaker cells, commonly found in dopaminergic neurons [103,104,105]. It is also unclear how the slow, massive waves of Ca^2+^ (oscillation readout) correlate with electrophysiological changes. One approach to this may be a double recording of Ca^2+^ and electrical signals. Excellent examples of this technology have been presented [98,106], but it requires extensive training and experience. As we are studying the coordination of cells over long distances, recording from single cells may need to be supplemented by the use of voltage sensors, such that information on several distant cells can be obtained [98]. The implementation is out of the scope of the present study, but it appears attractive for a follow-up. Our initial data suggest that the Ca^2+^ waves (with a wavelength of seconds) are distinct from APs in LUHMES cells (ms range), not just concerning the time domains, but also the overall pattern. A full quantification will require substantial additional experimental efforts.

Gap junctions are found in various brain regions and between different cell types of the human nervous system [107,108]. They are typically composed of hexamers, called connexins (Cx), that are expressed in the cell membrane of neighboring cells and form the gap junctions upon alignment [109,110]. Transcriptomic data show that Cx43 is highly expressed in LUHMES cells. Cx43 can be found between astrocytes [111] and in the cortex during neurodevelopment, and is thought to play a key role in the regulation of neurogenesis [112].

Gap junctional coupling and the initiation of synchronized Ca^2+^ waves have been shown to be strongly influenced by the ionic composition of the extracellular environment [113,114,115,116]. Blockade of gap junctions in vivo provides neuroprotection after perinatal global ischemia. Notably, elevated extracellular potassium levels, such as those observed during epileptic activity, can enhance gap junctional conductance [113,114]. Conversely, an increase in intracellular Ca^2+^ levels has been shown to result in gap junction uncoupling [115,116], which is prevented in the presence of high extracellular potassium (e.g., concentration of about 90 mM) [115,116].

In our study, extracellular Ca^2+^ was elevated from 2.5 mM to 3.8 mM. We speculate that the higher extracellular Ca^2+^ concentration does not necessarily lead to strongly increased intracellular Ca^2+^ concentrations, as these would have resulted in cell death. We favor the hypothesis that the conductivity of receptors and transporters in the cell membrane was tweaked by the increased calcium levels in a way that favored oscillations. However, detailed electrophysiological studies are required to obtain clear evidence for this.

Including the early onset of oscillatory activity at d2 and dye-transfer experiments showing spread to more than ten cells at d4, but only to one or two cells by d9, our data suggest that electrical synapses are more prominent during early stages of differentiation. Notably, in the human fetal cortex (17–23 gestational weeks), spontaneous depolarizations in subplate neurons were attenuated by octanol, CBX, and flufenamic acid, while gap-junction–permeable dyes did not spread to neighboring cells—pointing to a predominant hemichannel contribution at that developmental stage [95]. In contrast, LUHMES cultures show robust intercellular dye transfer and abundant Cx43 expression during early differentiation, consistent with bona fide gap-junctional coupling as the coordinating element in our system. Both diminish as maturation progresses, but are still well-retained on d9. Gene expression analysis and immunohistochemistry staining also revealed that gap junction occurrence was higher during early development stages and declined with neuronal maturation. This observation aligns with the theory proposed initially by Fischbach in 1972, which posits that electrical synapses contribute to the initial formation of neuronal circuits during neurodevelopment prior to the emergence of chemical synapses [16,82,83]. This is supported by findings in which coordinated Ca^2+^ transients in neighboring cells could be observed only in neuronal precursor cells of the developing brain and not in later stages [117]. Our data indicate that LUHMES cultures offer a unique opportunity to form electrically coupled neuronal networks as observed during neurodevelopment in the human brain. This may allow the development of a toxicological new approach methodology (NAM) for developmental neurotoxicity testing that is complementary to already existing methods. Presently available NAMs of the developmental neurotoxicity test battery (as described by a consensus of international stakeholders [118,119]) focus on morphological endpoints and electrical changes but may fail to detect toxicant actions on electrical synapses. 

The lack of models, knowledge, and experience with factors that modulate electrically coupled neuronal networks is a serious concern for research on developmental neurotoxicants. Without suitable model systems, it is difficult to identify potential toxicants that trigger developmental neurotoxicity (DNT) by disturbing electrical synapses. Failing to identify such toxicants would increase the risk for the human population to be exposed to such harmful agents, with potential consequences for an increased liability to develop neuropsychiatric and neurological disorders. This situation has been referred to as the silent pandemic of neurodevelopmental toxicity [120].

Gap junctional communication not only plays a role in neurodevelopment but also in several pathologies, ranging from brain tumors [121,122] to stroke-like conditions [123,124]. The oxygen-glucose deprivation (OGD) typically caused by a stroke leads to increased electrical coupling of neuronal cells and results eventually in neuronal cell death [113]. The gap junction blocker CBX has been shown to exhibit neuroprotective features during ischemic or hypoxic conditions, indicating that electrical synapses play a role in cell death mechanisms after OGD [113]. Another study using a trauma model, in which severe neuronal injury was induced by dropping a weight onto the neurons, also led to an increase in electrical connectivity [125]. The authors could demonstrate that cell death could partially be protected by blocking gap junctions. This indicates that gap junctions are involved in cell death signaling, possibly by spreading stress factors from cell to cell via electrical synapses [125], underscoring the need for reliable in vitro systems to study gap junctional communication. Thus, in addition to establishing neurodevelopmental toxicity assays, the LUHMES cell model could also have potential in studying brain injury mechanisms after stroke or trauma.

The importance of developing an in vitro gap junction functional assay, especially in a human cell model, is further accentuated by the fact that a variety of connexin knockouts in mice are lethal or cause severe impairments [126,127,128]. Moreover, in humans, more than 30 inherited connexin-linked diseases have been identified that range from motor-sensory neuropathy, e.g., X-linked Charcot–Marie–Tooth disease [129,130], to deafness [18,19,20], skin defects [131,132], and eye disorders [133,134].

In summary, we propose that the LUHMES cell model mimics the highly complex neuronal developmental process in which gap junction communication is the first neuronal circuit to form a functional network before chemical synapses take over. This may be a suitable model for studying this process without relying on animal models. It further opens the possibility to study DNT mechanisms, but also possibly neuronal injuries in which gap junctional communication also plays a role.

## Figures and Tables

**Figure 1 cells-14-01744-f001:**
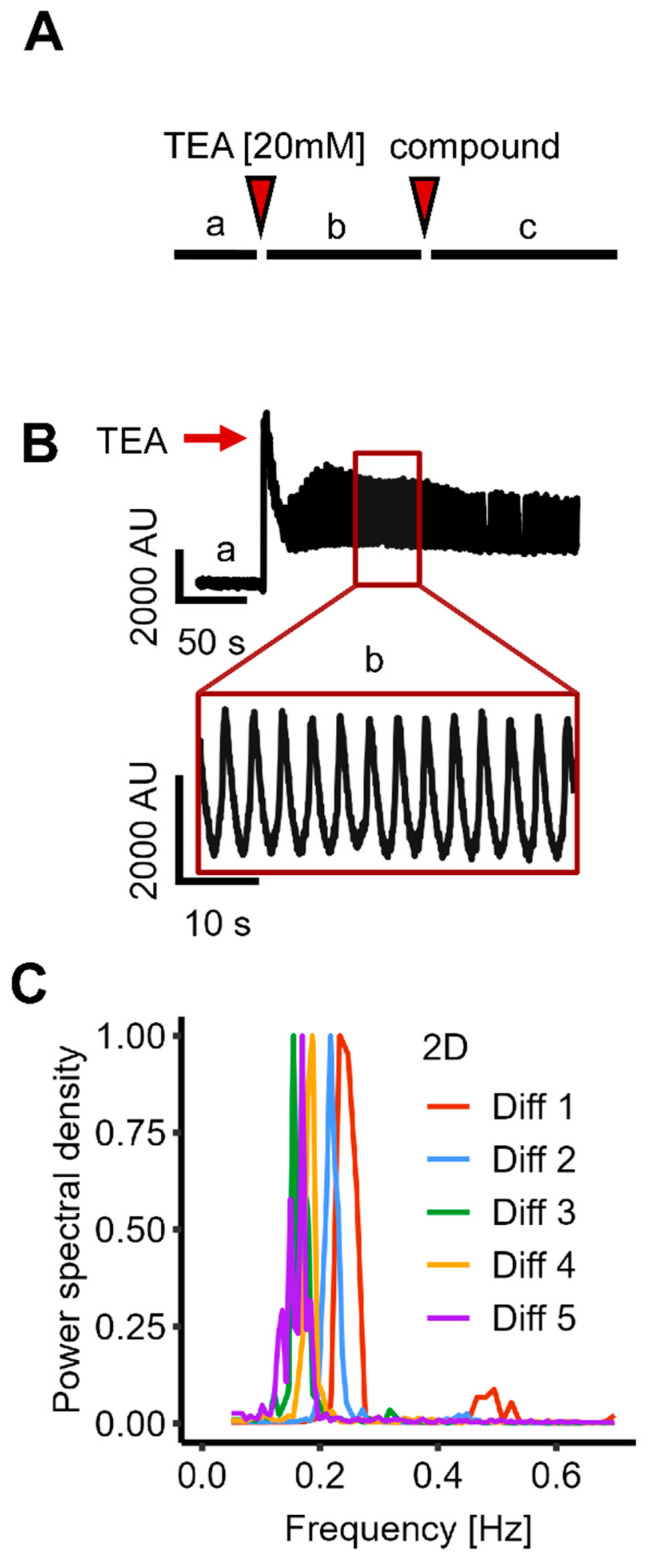
Triggering synchronized Ca^2+^ oscillations in conventional (2D) cultures of LUHMES neuronal-like cells at the network level. LUHMES cells were plated in 384-well plates and differentiated for a total time of nine days (d9 cultures). The differentiation medium was exchanged for oscillation medium, and cells were loaded with a Ca^2+^ indicator dye. Compounds were added by a robotic device. The total fluorescence of the entire well area was recorded (simultaneously for all 384 wells) continuously over time (one data point per well per time point). (**A**) Schematic representation of the recording protocol: (a) A two-minute baseline was recorded prior to the addition of TEA [20 mM]. (b,c) In compound screening experiments, test substances (or solvent controls) were added (c) five min after TEA. (**B**) Exemplary fluorescence trace (time on x-axis), showing a stable baseline, followed by robust oscillations induced by TEA. (**C**) A fast Fourier transformation (FFT) was performed on recordings from five independent experiments (Diff1-5) at d9 cells to analyze the oscillation frequency distribution. The FFT for each experiment shows an average of four wells per experiment (*n* = 20).

**Figure 2 cells-14-01744-f002:**
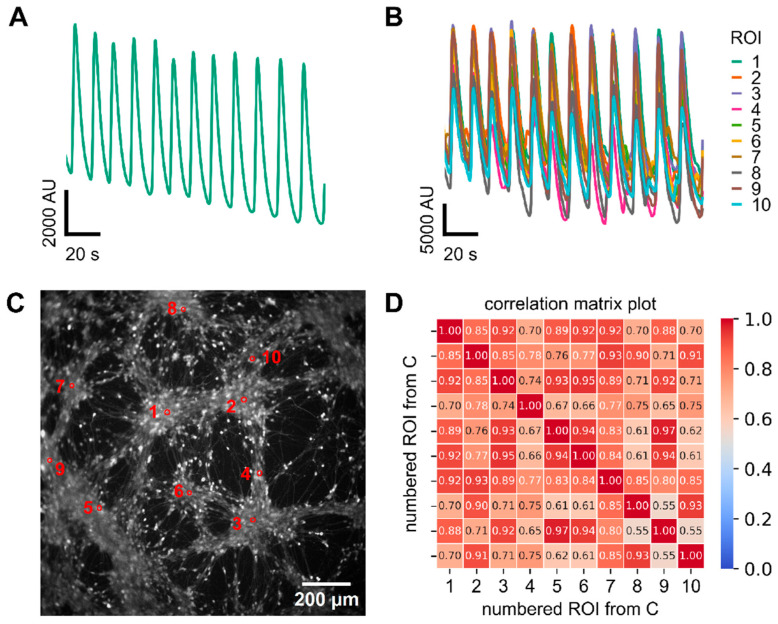
Assessment of synchronized Ca^2+^ oscillation in 2D-cultured LUHMES at the single cell level. Cells were treated as in Figure 1 (no test compound addition). (**A**) shows a fluorescence trace of an entire well with synchronized oscillatory activity. (**B**) Superimposed fluorescence traces from ten randomly selected regions of interest (ROIs). (**C**) Representative image of a 2D culture. The marked ROIs correspond to the ten cells shown in panel B. (**D**) The ten selected ROIs from panel C were used to generate a correlation matrix representing pairwise activity comparisons. This matrix served as the basis for calculating the global synchronization index (GSI), which yielded a value of 0.84 ± 0.02 for the 2D cultures when it was calculated from nine different experiments. Note: The apparent width of Ca^2+^ transients reflects calcium handling and indicator kinetics (and spatial averaging for whole-field traces) and is not interpreted as the time course of the underlying electrical events.

**Figure 3 cells-14-01744-f003:**
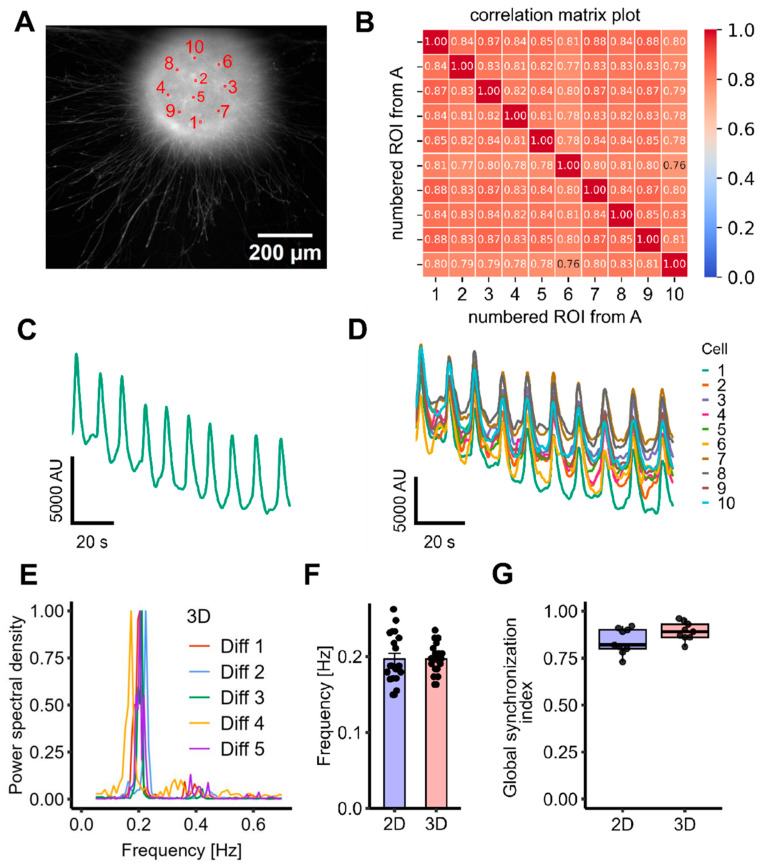
Correlation and frequency analysis reveal globally synchronized Ca^2+^ oscillations in LUHMES 3D networks, in comparison with 2D cultivation. LUHMES cells were grown as floating spheroids and then plated into imaging wells. Experiments were started, and data were recorded 48 h later. (**A**) Representative image of a spheroid stained with 2 µM of the calcium indicator dye Cal-520, AM. Ten randomly selected ROIs that were used for analysis are highlighted. (**B**) The ten selected ROIs from panel A were used to generate a correlation matrix representing pairwise activity comparisons. This matrix served as the basis for calculating the GSI. (**C**) Fluorescence trace based on data from whole-well imaging (i.e., recording fluorescence from the entire spheroid). (**D**) Superimposed fluorescence traces from ten randomly selected ROIs within the spheroid. (**E**) An FFT was performed on recordings from four wells per differentiation across five independent experiments (Diff 1–5) at d9 cells to determine oscillation frequency (*n* = 20). The FFT for each experiment shows an average of four wells per experiment. (**F**) The 3D spheroids exhibited a mean oscillation frequency of 0.200 ± 0.004 Hz (*n* = 20), comparable to that of 2D cultures (0.200 ± 0.007 Hz, *n* = 20). The difference was not statistically significant (Welch two-sample *t*-Test: t = −0.0062, *p* = 0.9951; Wilcoxon rank sum test: *p* = 0.7045). (**G**) The GSI was 0.89 ± 0.02 for 3D spheroids (*n* = 9, mean ± SEM) and thus did not differ from 2D cultures (see Figure 2). The difference was not statistically significant (Welch two-sample *t*-Test: t = −1.8937, *p* = 0.0783; Wilcoxon rank sum test: *p* = 0.1443).

**Figure 4 cells-14-01744-f004:**
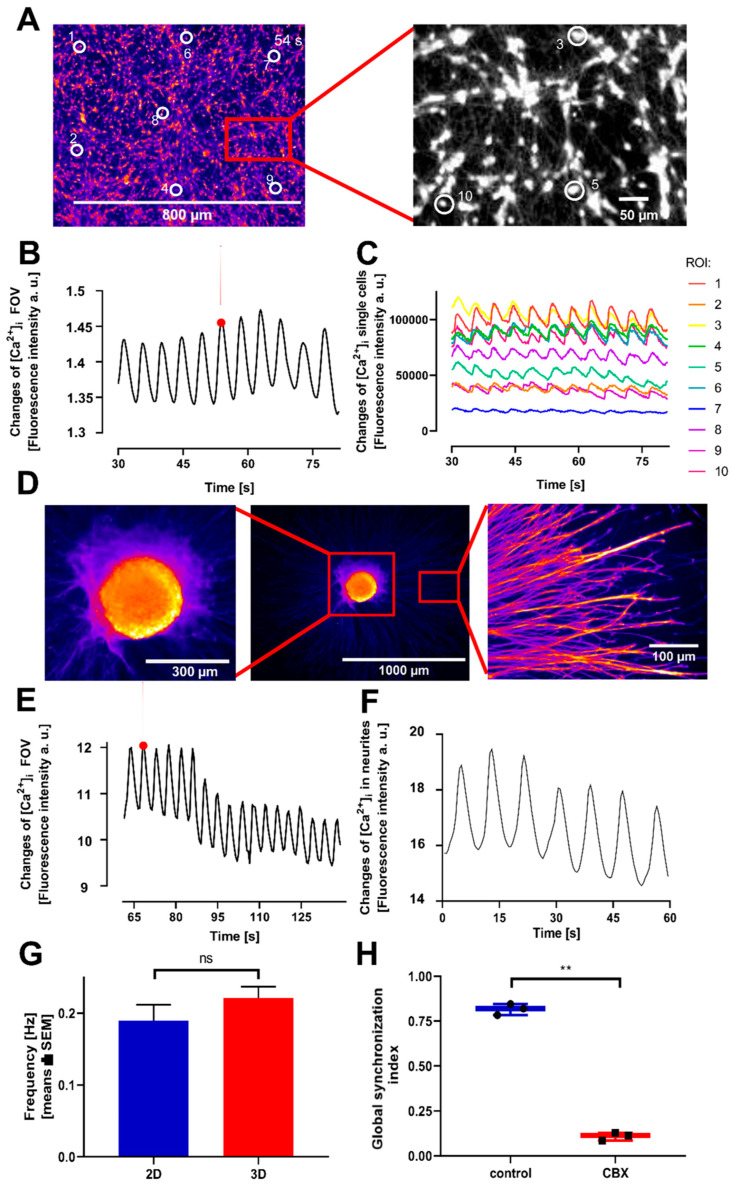
Inter-laboratory transferability of the assay for coordinated Ca^2+^ oscillations in neuronal-like cultures. LUHMES neurons were prepared for recording as in Figure 1 and Figure 3. (**A**) Representative fluorescent image (Cal520 indicator) of a 2D neuronal-like cell culture, taken at the peak of an oscillation. Ten regions of interest (ROIs), at distances up to 900 µm, are shown. Note that some ROIs are within the magnified insert and represent single cells. (**B**) Fluorescence traces of the entire field of view (FOV), as indicated in (**A**). (**C**) Superimposed fluorescence traces from ten randomly selected ROIs within the FOV shown in (**A**,**B**). (**D**) Middle image: Representative image of a 3D cell culture at the peak of an oscillation. A magnification is shown to the left. Right image: magnification of the spheroid’s neurite area. (**E**) Fluorescence traces of the entire spheroid. (**F**) Fluorescence trace of the neurites in the selected field (on the right side of (**A**)), showing synchronized oscillatory activity. (**G**) Comparison of the mean frequency between 2D (0.19 ± 0.022 Hz) and 3D (0.22 ± 0.016 Hz). There was no significant difference (ns = not significant). (**H**) Comparison of the GSI of untreated 2D LUHMES cells (control; GSI = 0.82 ± 0.018%) and CBX (100 µM)-treated cells (GSI = 0.11 ± 0.012%). The difference was significant (Paired two-tailed *t*-test: *p*-value = 0.0013, ** = significant). The representative correlation matrix plots are shown in Appendix A.

**Figure 5 cells-14-01744-f005:**
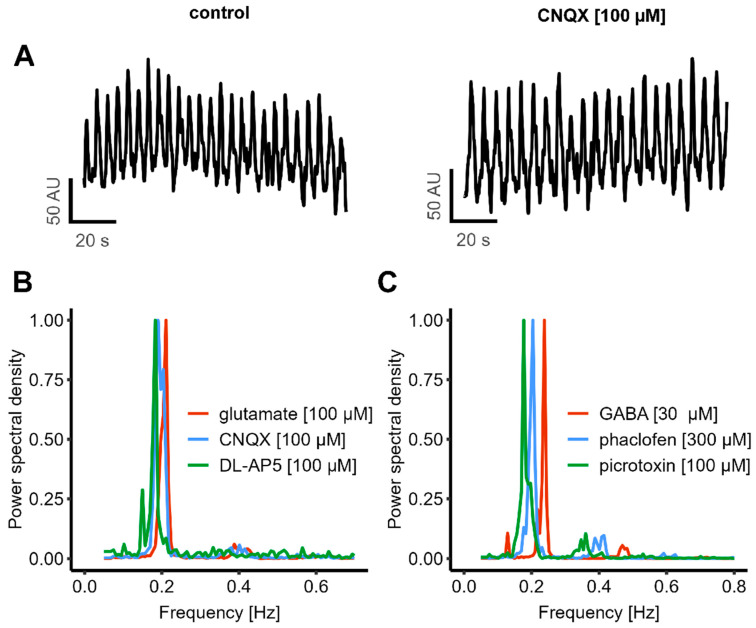
Mechanistic investigation of oscillatory activity using known synaptic modulators.Cells were prepared for the experiments as shown in Figure 1. (**A**) CNQX, an AMPA receptor antagonist, did not alter the oscillation frequency. (**B**) Pharmacological inhibition of the glutamatergic system was performed using CNQX (0.190 ± 0.008 Hz) and DL-AP5 (NMDA receptor blocker, 0.180 ± 0.002 Hz). Additionally, glutamate was exogenously applied (oscillation frequency: 0.200 ± 0.006). (**C**) The role of the GABAergic system was examined by applying picrotoxin (GABAA receptor antagonist, 0.190 ± 0.006 Hz) and phaclofen (GABAB receptor antagonist, 0.190 ± 0.009 Hz). Additionally, GABA was added (oscillation frequency: 0.220 ± 0.007 Hz). None of the treatments had a significant effect on the oscillation frequency. All data represent mean ± SEM, *n* = 3.

**Figure 6 cells-14-01744-f006:**
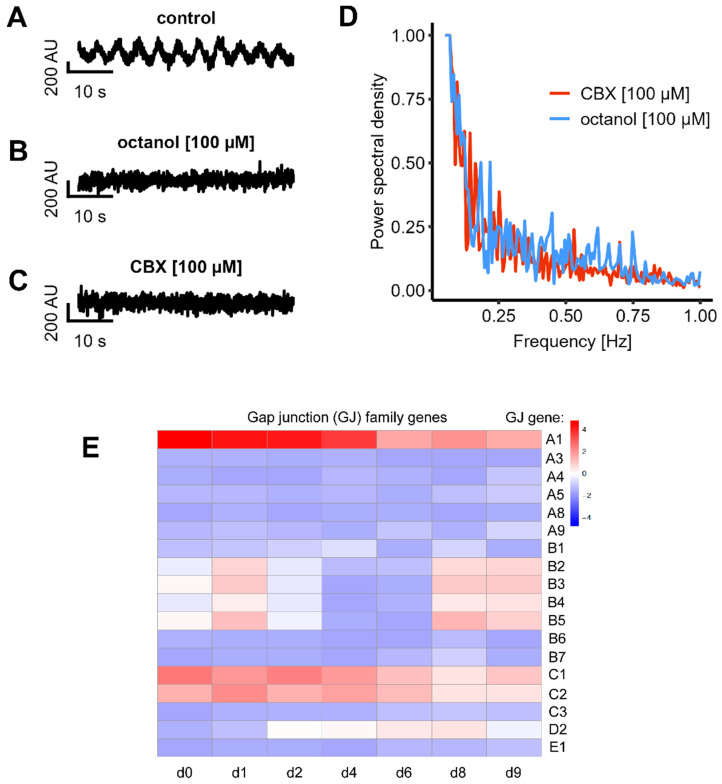
Characterization of gap junctions in LUHMES cultures. Cells were prepared for the experiments as shown in Figure 1. (**A**–**C**) The inhibition of gap junctions was performed with CBX and octanol. Representative oscillation traces are shown. (**D**) FFT of the fluorescence signal following CBX (*n* = 4) and Octanol (*n* = 4) treatment: no clear frequency peaks were detectable. (**E**) The expression level of genes coding for gap junction proteins was quantified by TempOSeq analysis. Data are presented in the form of a heat map for different differentiation days of LUHMES (x-axis). The gap junction (GJ) gene GJA1 (encoding connexin 43) is shown in the top line. Pink to red colors indicate expression levels of 4–16 counts per million reads (log2-scaled coding). The expression pattern for GJC1 and GJC2 was similar, with the highest levels in immature cells and a gradual decline (the full original data set is found in the Appendix A).

**Figure 7 cells-14-01744-f007:**
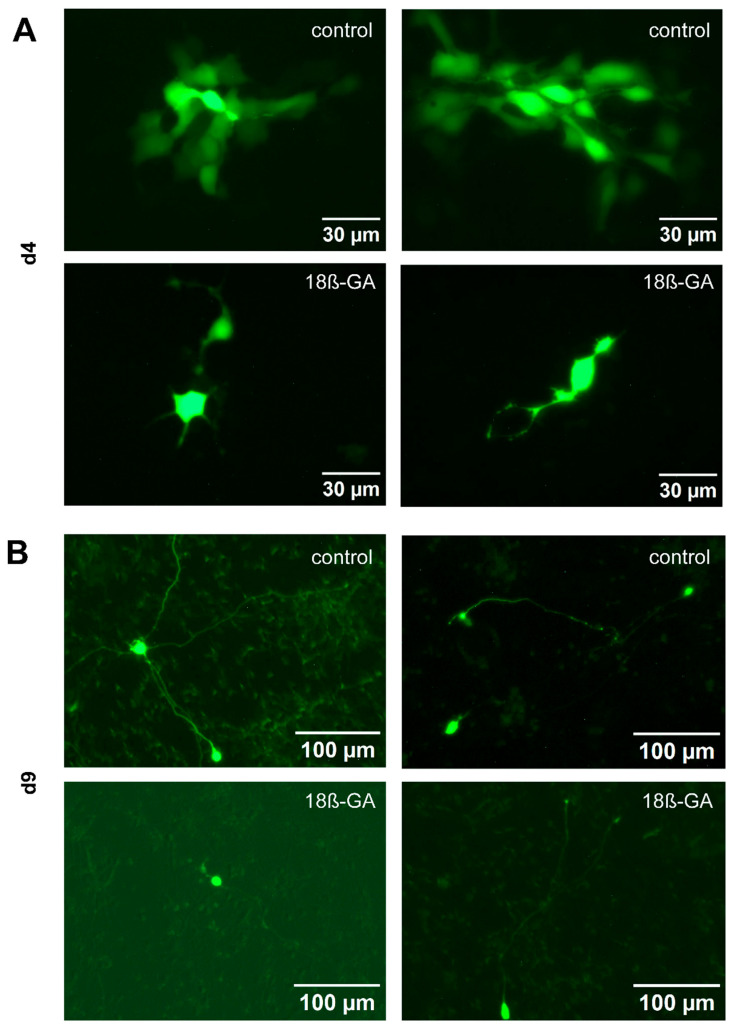
Dye transfer experiments using patch-clamp to demonstrate functional gap junction coupling in LUHMES cells. Alexa-488 was used to assess intercellular dye transfer at two differentiation stages. (**A**) At d4, the dye diffused to more than ten neighboring cells under control conditions (*n* = 6), indicating robust gap junction connectivity. Following application of the gap junction blocker 18β-GA [30 µM], dye transfer was either completely inhibited or limited to a single adjacent cell (*n* = 6). (**B**) At d9, under control conditions, dye spread was restricted to only one or two neighboring cells (*n* = 6). In the presence of 18β-GA [30 µM], the dye remained confined to the patched cell (*n* = 6).

**Figure 8 cells-14-01744-f008:**
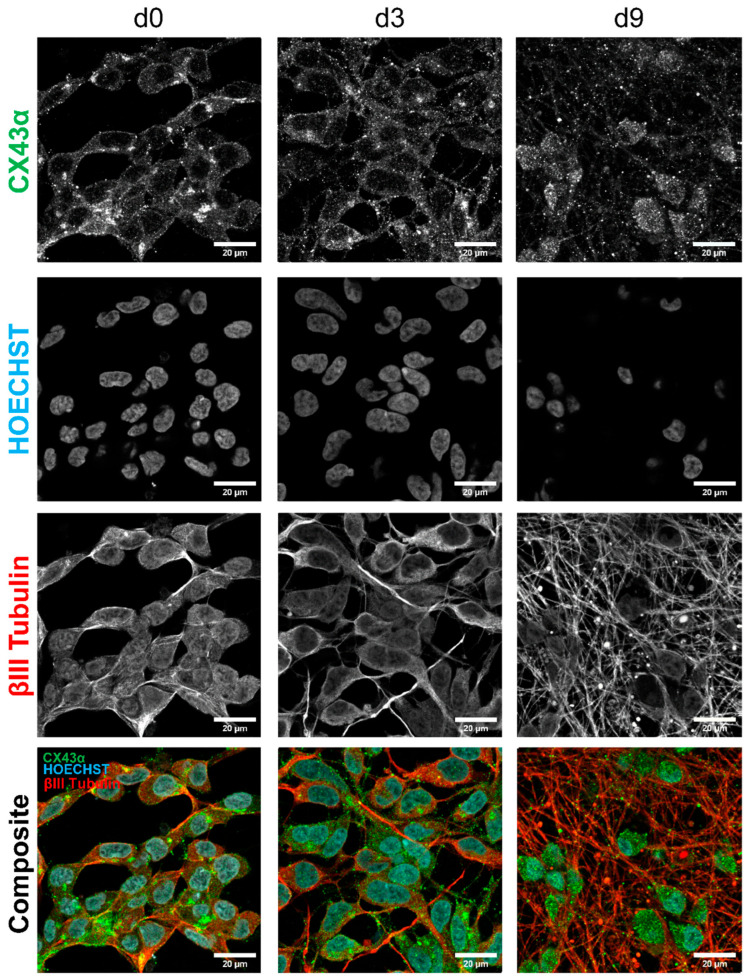
Immunohistochemistry staining for the expression of gap junction protein connexin 43α, ßIII-Tublin, and Hoechst in LUHMES cells. LUHMES cultures were differentiated for 0, 3, or 9 days on ibidi eight-well slides, respectively. The LUHMES cells were then fixed and stained with antibodies against connexin 43α, ßIII-tubulin, and Hoechst H-33342 as a nuclear stain and imaged using a confocal microscope. Connexin 43α staining is shown in green, ßIII-tubulin staining in red, and Hoechst H-33342 in blue.

**Figure 9 cells-14-01744-f009:**
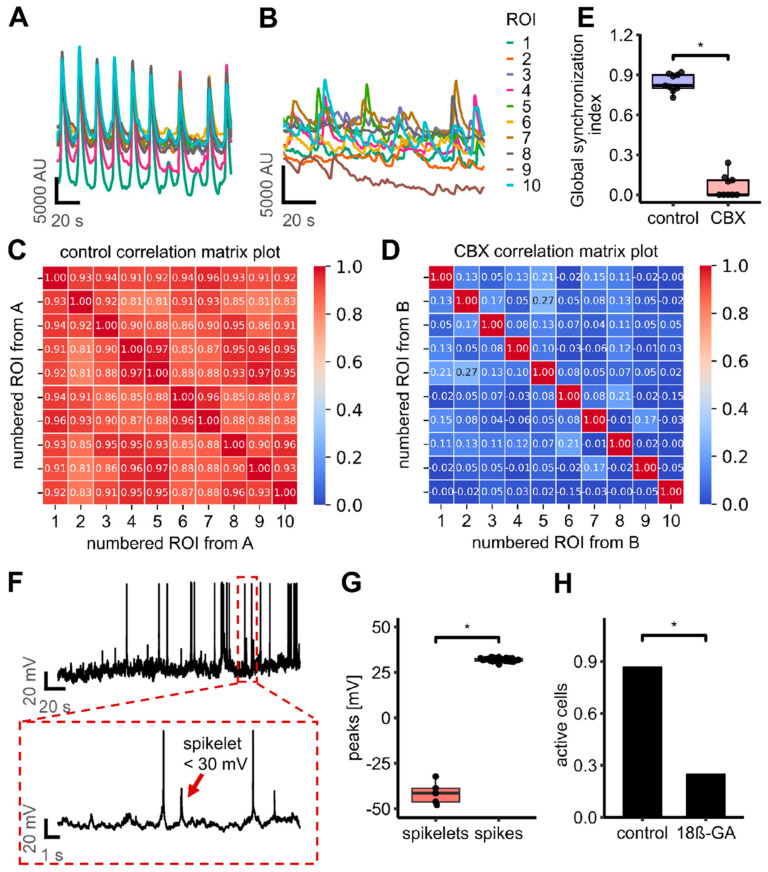
Analysis of the contribution of gap junctions to Ca^2+^ oscillations in LUHMES cultures. (**A**,**B**) Superimposed fluorescence traces from ten randomly selected ROIs, recorded using a high-resolution microscope, reveal synchronized oscillatory activity under control conditions (**A**). Following the application of the gap junction blocker CBX, oscillatory activity persists at the single-cell level, but synchrony is lost (**B**). (**C**,**D**) The corresponding correlation matrices indicate a highly synchronized activity under control conditions (**C**) but very low to no synchronized activity in the presence of CBX (**D**). (**E**) The GSI under control conditions was 0.84 ± 0.02 (*n* = 9). After CBX treatment, the GSI decreases significantly to 0.06 ± 0.03 (*n* = 9) (Welch two-sample *t*-Test: t = 21.326, *p* = 1.092× 10−12; Wilcoxon rank sum test: *p* = 0.0004). (**F**) Whole-cell patch-clamp recordings from LUHMES d9 cells under oscillatory conditions revealed small (amplitude < 30 mV) depolarization events (“spikelets”, red arrow), characteristic of gap junction-mediated electrical coupling. (**G**) Spikes and spikelets can be differentiated into two distinct groups. The difference was significant (Wilcoxon rank sum test: *p*-value = 0.0002). (**H**) Overall activity was decreased after the application of the gap junction blocker 18β-GA. 25% of the cells still showed spontaneous activity. The difference was significant (Wilcoxon rank sum test: *p*-value = 0.0016). A star (*) represents a significant difference.

## Data Availability

All data presented in manuscript figures are available in Excel files in the supplement, such that other displays or statistical approaches may be applied to them.

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
