# Peer review of "Gap Junctional Communication Required for the Establishment of Long-Term Robust Ca^2+^ Oscillations Across Human Neuronal Spheroids and Extended 2D Cultures"

_cells, 2025, doi:10.3390/cells14211744_

Round 1
Reviewer 1 Report
Comments and Suggestions for Authors
In this study, the authors used human dopaminergic LUHMES cells to establish a protocol for generating reproducible intracellular Ca²⁺ oscillations in both two-dimensional and three-dimensional cultures. Oscillatory activity was induced through ionic manipulation and potassium channel blockade. The reported oscillations exhibited extremely stable and slow frequencies, with a period of approximately 20 seconds—each calcium peak requiring 20 full seconds to recur. The authors further claim high synchronization indices across millimeter-scale cultures, and that these features were consistently reproduced in independent experiments conducted in two laboratories.
The experiments are technically well executed, and the overall study is of high methodological quality. However, the authors must revise their terminology throughout the manuscript. The repeated use of the term “neuron” is not justified in many contexts, as the presented data do not support that the cells in question are functionally or morphologically differentiated neurons.
In summary, this is a technically competent study with clear experimental execution. However, the authors’ interpretation and terminology are somewhat misleading. The data do not support the claim that the recorded cells are differentiated neurons. The observed calcium oscillations are characteristic of metabolic activity in undifferentiated progenitors, not neuronal electrical activity. Substantial revisions of terminology are required before the manuscript can be published.
1.
The suggested link between small depolarizations (Fig. 9F) and super-large Ca transients was not established with simultaneous whole-cell and Ca imaging recording, and thus is weak. Examples of dual whole-cell & Ca imaging:
PMID: 24157591
PMID: 39367010
2.
Definition of a Neuron: A neuron is a differentiated cell characterized by 5 cardinal features:
1a. An elaborate dendritic arbor.
1b. Functional synaptic connections.
1c. Generation of full-amplitude, fast sodium-dependent action potentials.
1d. In calcium imaging, neuronal action potentials appear as sharp, transient Ca²⁺ signals (<1 second in duration).
1e. In contrast, spontaneous neuronal activity in culture is irregular and not uniformly periodic.
The experimental results presented here are inconsistent with the DEFINITION OF A NEURON (items 1a to 1e), indicating that the recorded cells are undifferentiated.
Therefore, the term “neuron” should be carefully reconsidered, or replaced with something more appropriate, anywhere in the text.
3.
The Ca²⁺ transients shown in Fig. 2 are inconsistent with neuronal activity. Single-cell imaging traces suggest that these are not neurons. The authors should consult the literature on spontaneous calcium transients in developing neurons. These typically differ dramatically from the signals observed in this study.
4.
Calcium transients with a half-amplitude duration of ~10 seconds (Fig. 2) are not neuronal signals. Such long-duration events are metabolic Ca²⁺ oscillations, commonly found in non-differentiated or metabolically active cells. Transients that require 10 seconds to rise from trough to peak cannot be attributed to neuronal electrical activity.
5.
Young neurons in culture do not require TEA (a K⁺ channel blocker) to generate spontaneous calcium transients. In this study, the cells exhibit no spontaneous activity before TEA application (Fig. 1), further supporting that they are not differentiated neurons and should not be described as such.
6.
At line 255, the authors state: “We assumed that neurons in their resting state, or non-synchronized cultures of many neurons (about 20,000 cells/well), would produce an average Ca²⁺ signal that does not change in a regular (oscillating) way over time.”
However, neural progenitors are known to exhibit rhythmic Ca²⁺ oscillations during early neurogenesis, often appearing in immature, undifferentiated cells around day 9 in culture. This likely explains the findings in Fig. 1. The authors have not provided convincing evidence of neuronal differentiation in their wells. As such, their conclusions are scientifically unsupported and do not meet the rigor required for publication.
7.
CNQX and DL-AP5, which block glutamatergic synaptic transmission, did not affect the observed oscillations This is consistent with STRONG activity originating from non-neuronal progenitors (it is not coming from differentiated neurons).
The dominant cell type in this culture is therefore undifferentiated.
As noted earlier, a true neuron is defined by the presence of mature morphology, synapses, and action potential firing manifested as rapid, sharp (<1 s) Ca²⁺ spikes, that occur IREGULARLY over time. In contrast, the super slow, metabolic Ca²⁺ signals presented in the current manuscript, originate from non-differentiated cells lacking synaptic connectivity.
8.
(Line 17): “In this study, we explored the usefulness of LUHMES cultures to establish a novel model of Ca²⁺ oscillations in human neurons.”
What exactly is “novel” here?
The study does not present a novel concept based on at least 5 findings.
7-1. LUHMES cultures have been used for >10 years.
7-2. LUHMES cells were cultured following established protocols (19, 31, 32).
7-3. Ca imaging in culture, or in developing brain tissue, has been used for >10 years.
7-4. Three-dimensional spheroids have been used for >10 years.
7-5. Ca2+ oscillations have been observed in LUHMES cells under pharmacological stimulations.
The authors must clearly state what aspect of this work is genuinely novel. Please spell it out in a paragraph devoted to "novelty".
9.
The reagent is incorrectly identified as “Cal-520™ (AAT Bioquest, Pleasanton, USA).”
The correct reagent used was likely Cal-520, AM (AAT Bioquest, Pleasanton, CA, USA).
Note that Cal-520 membrane-impermeable forms, are available.
10.
Fig. 8 does not confirm that the dominant cell type is neuronal. βIII-tubulin is not neuron-specific. It is expressed in neural progenitors and neuroblasts prior to full differentiation and is also upregulated in several carcinoma cell types. The detection of βIII-tubulin at Day 0 (Fig. 8) strongly suggests that the cells are not differentiated neurons, and should not be labeled as such in the manuscript.
11.
The age of the cells is not indicated in Fig. 9F, where occasional action potentials were observed. It is possible that a small subset of cells became electrically active, but the majority are clearly not differentiated neurons. The culture appears dominated by undifferentiated cells generating very slow Ca²⁺ oscillations with a 20-second period.
12.
“Red arrow” in Fig. 9F is not explained in the figure caption.
13.
The authors have missed the study conducted in developing human neurons, developing human brains, brain slices, titled: “Connexin hemichannels contribute to spontaneous electrical activity in the human fetal cortex”. PMID: 25197082
>>> Similarities between PMID: 25197082 and the current Kormann et al (2025) study:
14a. Human cells.
14b. Whole-cell recordings of spontaneous activity. Related to Fig. 9F in Kormann et al (2025).
14c. Blockers of synaptic transmission (glutamate, GABA, etc.) do not change spontaneous neuronal activity. Related to Kormann’s Fig. 5.
14d. Blockers of gap junctions DO change spontaneous neuronal activity. Related to Kormann’s Fig. 6.
14e. Immunolabeling revealed expression of several connexins in human neurons. Related to Kormann’s Fig. 8.
Srdjan D. Antic, MD, UConn Health, Farmington, CT, USA
Reviewer 2 Report
Comments and Suggestions for Authors
The present manuscript by Kormann et al. entitled "Gap junctional communication required for the establishment of long-term robust Ca2+ oscillations across human neuronal spheroids and extended 2D cultures." demonstrates the fundamental networking and intracellular Ca²⁺ oscillatory characteristics of cultured human dopaminergic LUHMES cell line in two-dimensional monolayers and three-dimensional spheroids. Authors used single cell imaging and pharmacological interventions using gap junction blockers to show reproducible high synchronization indices across millimeter-scale cultures in this cell line. There are few queries/comments that need to be addressed as follows:
Query#1. In methods section, authors mentioned the reference for LUHMES cell culture protocol, however it would be important to mention brief details about harvesting/deriving these cell lines.
Query#2. In fig.1, authors used 20mM of TEA concentration to stimulate intracellular Ca²⁺ oscillations in 2D cultured LUHMES cell line, which is relatively higher millimolar range for human dopaminergic neuronal cells line, did authors conduct a dose response curve for EC50 conc. of TEA? Also, authors used high extracellular calcium i.e. 3.8 mM, that might induce neurotoxic effect on LUHMES cells, authors need to show the dose response curve for toxicity of both TEA and extracellular calcium.
Query#3. In fig.1c, authors need to elaborate Diff1-5, as its not cleared in the figure. Do they refer to differentiation day 1-5? Also, a bar graph representation of statistical analysis could better elaborate the comparison in frequencies.
Query#4. In fig.2, authors need to show the quantification of fluorescence and representative bar graph for any statistical difference.
Query#5. In fig. 4H, GSI index of control vs CBX looks significantly different, is this statistically significant? Authors need to show statistical analysis with p value and asterisk.
Query#6. In fig. 5, authors did not observe any statistical difference in oscillatory frequency after treatment with synaptic modulators (Glutamate and GABA) n=3. Authors need to show bar graph or curve to see if there is any trend.
Query#7. In fig. 6, authors showed declination in GJC1 and GJC2 gene expression of cultures day in vitro (d1-9), immature neurons showing highest expression of both genes. Did author conduct gap junction blocker treatments at immature vs mature cultures? It seems like d2-4 is optimal day in vitro for this assay. Why did authors not show d3 data in this study?
Query#8. In fig. 9g and 9h, authors need to show statistical p values and asterisk on the bar graph.
Query#9. Authors studied oscillatory calcium response in human dopaminergic neuronal model (LUHMES cell line), however cortical neuronal model such as HCN-2 and hippocampal neuronal model HT22 have been vastly studied synaptic plasticity and signaling pathways. Why did authors choose LUHMES cell line? to validate the specificity of this cell line towards oscillatory calcium flux, authors need to compare abovementioned cell line.
Reviewer 3 Report
Comments and Suggestions for Authors
Kormann et al. characterize intracellular Ca2+ oscillations occurring in two-dimensional monolayers and three-dimensional spheroids obtained by using LUHMES neurons (Figs. 1-4) and examine cellular mechanisms for the production of the oscillations (Figs. 5-9). The experiments were preformed by using Ca2+ imaging, pharmacological, transcriptomic, dye-transfer, immunohistochemical and whole-cell patch-clamp methods. As a result, it was revealed that the Ca2+ oscillations are produced by gap junctions but not glutamatergic and GABAergic transmissions. This manuscript is written by using many abbreviations that are not commonly used (for example, “ROI”, “NMI”, “GSI”, “NAM” and “DNT”), and so it is somewhat difficult to read. Some of the drugs used are not listed in Methods, and so their source is unknown. There are many ambiguities in English and scientific writing. This manuscript is not written with any care, because so many simple mistakes are noted. Several points, which should be addressed and may serve to amend this manuscript, are as follows:
- Line 40: not “(1)” but “[1]”. All reference numbers should be given by using [..] (see Instructions for Authors in Cells).
- Line 84: please delete “Missouri, USA” (see line 83). It is not necessary to repeatedly write the location of the company. Such a problem is apparent throughout this manuscript (for instance, see lines 87, 91, 95, 97, 102, 114, 160 and 197). This point should be amended.
- Line 88: it will be better to use “L” not “l” (see line 140). “L” should be used throughout this manuscript (for example, in lines 166 and 167).
- Line 121: not “1h” but “1 h”. A similar problem is seen in line 159. A space should be put between value and unit throughout the manuscript.
- Line 163: “from (19)” is an awkward expression. A similar problem is seen in lines 215 and 555. This point should be amended.
- Line 171: please use either “min” (here; also in line 194) or “minutes” (see line 160) throughout this manuscript.
- Line 172: “IHC” should be spelled out.
- Line 173: “FCS” should be spelled out.
- Third and fourth paragraphs on page 4: these paragraphs should state from which company the drugs and equipment used were purchased and where that company is located.
- Line 226: generally, not “fourier” but “Fourier” (see line 644). A similar problem is seen on lines 228, 282 and 336. This point should be amended.
- Line 229: not “work” but “works”? Please check English language.
- Line 250: how was the osmolality of the solution adjusted when TEA (20 mM) was added? From which company TEA was purchased? These points should be made clear.
- Line 253: please use either “FDSS/μCELL” (here) or “FDSS/μCell” (see line 117) throughout this manuscript.
- Lines 260, 266 and 271: the way the “Fig” is displayed is different in each case. Please amend this point.
- Lines 270 and 271: the “n” value that gives the variability of the data should be stated.
- Line 291: “ROI” should be given in an abbreviation list.
- Line 302: as different from the figure legend title given here, those of Figs. 3-9 are written in italic and underline. This point should be amended.
- Lines 320 and 325: the “n” value that gives the variability of the data should be stated.
- 3G: is there a problem with the “[%]” on the vertical axis? Please check this point.
- Line 328: this title does not adequately reflect the content of Fig. 3. This point should be amended.
- Line 331: please use either “Cal-520” (here) or “Cal-520TM” (see line 119).
- Line 341: “mean ± SEM” should be deleted (see line 236).
- Line 348: please spell out “NMI”.
- Line 352: each of Figs. 4A, B, C, D, E, F, G and H should be explained separately.
- 4H: is there a problem with the “[%]” on the vertical axis? Please check this point.
- Line 354: “2+” in “Ca2+” should be superscript.
- Line 355: not “Fig” but “Figs”? Please amend this point.
- Line 366: not “Carbenoxelone” but “Carbenoxolone” (see line 403)? Please check this point.
- Page 12: from which company each of carbenoxolone, CNQX, AP-5, glutamate, GABA, picrotoxin and phaclofen was purchased? Such an information should be given in Methods. Which is right, AP-5 or DL-AP-5 (see Fig. 5)? This point should be made clear.
- Page 12: do the LUHMES neurons used express AMPA, NMDA, GABAA and GABAB receptors? What about glycine receptors, although cultured central neurons are known to express glycine receptors? Did glutamate or GABA itself produce any response? Please reply to these questions.
- 5: in relation to this figure, what kind of synaptic responses are recorded when the whole-cell patch-clamp method is applied to the 2D cultured neurons used? Are spontaneous synaptic currents observed? Please reply to this question.
- Legend of Fig. 5 on page 13: all of “mean ± SEM” should be deleted.
- Line 429: is right the way of the presentation of “n” and “cell numbers”? Please check this point.
- Line 430: each of Figs. 7A and 7B should be explained separately.
- Lines 431 and 432: “18β-GA” should be spelled out in line 160, not here.
- Line 444 and 446: the concentration of 18β-GA used should be given here.
- Line 467: “Suppl. Fig. 3” should not appear after “Suppl. Fig. 4” (see line 367). Where are “Suppl. Fig. 1” and “Suppl. Fig. 2” in the text? Please make these points clear.
- Line 476: is “IBIDI” OK in writing? Please see line 165. Please make this point clear.
- Line 475: please use either “connexin 43α” or “connexin 43 alpha” (line 174).
- Line 478: why was “ßIII-Tubuling” used? From which company “ßIII-Tubuling” was obtained? Please make these points clear.
- Lines 476, 478 and 479: the way HOECHST is written is all different (also see line 179). This point should be amended.
- Line 492: the “n” value that gives the variability of the data should be stated.
- Line 497: the concentration of TEA used should be stated.
- Second paragraph on page 18: it appears that the patch-clamp technique is not being used effectively. Could not the patch-clamp technique be applied to two adjacent 2D cultured neurons simultaneously? This would serve to know whether there is direct electrical coupling. Please reply to this question.
- 9E and H: is there a problem with the “[%]” on the vertical axis? Please check this point.
- Lines 518 and 521: both “Carbenoxolone” and “CBX” should not be used in the same figure legend. This point should be amended.
- Line 521: “(D)” should be put following “CBX”.
- Lines 521 and 522: the “n” value that gives the variability of the data should be stated.
- Lines 523-525: there is no explanation about the graph of Fig. G. Please amend this point.
- Fourth paragraph on page 20: please explain in more detail how a change in “transmitter release at the presynaptic membrane” produced by K+-channel inhibition leads to “synchronized Ca2+ waves”. Although the authors seem to consider an involvement of “TEA-induced depolarization”, is this depolarization post- or presynaptic in origin? If this depolarization is postsynaptic, this possibility will be investigated by using the whole-cell patch-clamp technique. Please reply to this question.
- Line 604 and 612: “NAM” and “DNT” should be given in an abbreviation list.
- Lines 658 and 659: this sentence should be underlined and in italic.
- Line 665: where is “Suppl. Table 1” in the text? Please amend this point.
- Line 670: not “Table 1” but “Suppl. Table 1”? Please check this point.
- There seem to be more simple and scientific mistakes than pointed out above. This manuscript should be checked very carefully.
Round 2
Reviewer 2 Report
Comments and Suggestions for Authors
Authors have substantially revised the present manuscript and satisfactorily addressed all queries and suggestions.
Author Response
We thank the reviewer for helping to improve the manuscript.
Reviewer 3 Report
Comments and Suggestions for Authors
Many of the issues this reviewer pointed out have been addressed, but the following minor points should be considered:
- Line 25: not “.. coupling This was” but “.. coupling. This was”.
- Page 2: it may be better to give the list of abbreviations in an alphabetic order. All of the abbreviations used in this manuscript do not seem to be listed, for example, AP (line 69), LUHMES (line 125), PLO (line 132), PBS (line 138) and PEI (line 148).
- Line 69: not “action potentials (AP)” but “action potentials (APs)”.
- Line 101: please use “APs”.
- Lines 139 and 140: please use either “minute” or “min” throughout this manuscript.
- Line 142: is “6” in “N6” superscript? Please check this point.
- Line 204: not “))” but “)”.
- Line 209: not “1nA” but “1 nA”.
- Line 356: it is not necessary to repeatedly define “GSI” (see line 265). This definition is also given in the list of abbreviation.
- 3G, 4H, 9E and 9H: do values (such as 0-1) given in the vertical axes of these figures show percentage (%) values? Please check this point.
- Line 391: not “Cal-520” but “Cal-520 AM”? Please check this point.
- Line 454: “A,B” should be shown in bold. The same issue is seen in many other places.
- The legend of Fig. 4 on page 14: please give an explanation for “ns” and “**”.
- Lines 422, 423 and 429: please give the concentrations of oligomycin, Glutor and calcein-AM used.
- Line 431: what is “Fig. Sx”? Please make this point clear.
- Line 462: “Suppl. Fig. 2” should be shown in bold.
- Line 554: “Fig. 5E” cannot be found in this manuscript. Please amend this point.
- Line 562: not “cultures,” but “cultures.”.
- Lines 574 and 575: is “antibody against Hoechst H-33342” OK? Please check this point.
- Line 576: not “H33342in” but “H33342 in”.
- Line 588: please use “GSI”.
- Line 599: not “[20mM]” but “[20 mM]”.
- Lines 623 and 624: is “e-12” OK? Please check this point.
- Line 714: please use “APs”.
- Line 812: please give the concentration of calcein-AM used.
- Line 826: not “Fig. 2” but “Fig. 3”? Please check this point.
- Line 830: not “Fig. 3” but “Fig. 5”? Please check this point.
- Line 842: not “Fig. 5” but “Fig. 2”? Please check this point.
- References: the style of title writing is different between [21] and [22]. Please amend this point. All References should be checked in writing style.
- There may be more mistakes than pointed out above. Please check your manuscript very carefully.
Author Response
Thank you for your comments. We changed the suggestions in the manuscript accordingly. Regarding Ref#22, we double checked the format and it is correctly cited.